# Hunting Network Anomalies in a Railway Axle Counter System

**DOI:** 10.3390/s23063122

**Published:** 2023-03-14

**Authors:** Karel Kuchar, Eva Holasova, Ondrej Pospisil, Henri Ruotsalainen, Radek Fujdiak, Adrian Wagner

**Affiliations:** 1Department of Telecommunications, Faculty of Electrical Engineering and Communications, Brno University of Technology, Technicka 12, 616 00 Brno, Czech Republic; 2Institute of IT Security Research, St. Pölten University of Applied Sciences, Campus-Platz 1, 3100 St. Pölten, Austria; 3Department of Rail Technology & Mobility, Carl Ritter von Ghega Institute for Integrated Mobility Research, St. Pölten University of Applied Sciences, Campus-Platz 1, 3100 St. Pölten, Austria

**Keywords:** attack classification, axle counter, feature selection, ICS, neural network, OT, railway, testbed threat

## Abstract

This paper presents a comprehensive investigation of machine learning-based intrusion detection methods to reveal cyber attacks in railway axle counting networks. In contrast to the state-of-the-art works, our experimental results are validated with testbed-based real-world axle counting components. Furthermore, we aimed to detect targeted attacks on axle counting systems, which have higher impacts than conventional network attacks. We present a comprehensive investigation of machine learning-based intrusion detection methods to reveal cyber attacks in railway axle counting networks. According to our findings, the proposed machine learning-based models were able to categorize six different network states (normal and under attack). The overall accuracy of the initial models was ca. 70–100% for the test data set in laboratory conditions. In operational conditions, the accuracy decreased to under 50%. To increase the accuracy, we introduce a novel input data-preprocessing method with the denoted gamma parameter. This increased the accuracy of the deep neural network model to 69.52% for six labels, 85.11% for five labels, and 92.02% for two labels. The gamma parameter also removed the dependence on the time series, enabled relevant classification of data in the real network, and increased the accuracy of the model in real operations. This parameter is influenced by simulated attacks and, thus, allows the classification of traffic into specified classes.

## 1. Introduction

Railway transportation plays a leading role as one of the most environmentally friendly ways of traveling and logistics. According to [1], railways are 15 times more energy-efficient than aviation and, thus, their contribution to the EU’s Green Deal to reduce ${\mathrm{CO}}_{2}$ emissions by 90% until 2050 is noteworthy. Therefore, it can be safely assumed that in the near future, rail-based passenger transportation, as well as freight transportation, will grow compared to other means of transport. Similar to other industries, digital innovations that define Industry 4.0 have been introduced to railway infrastructure [2]. These innovations promote, for example, process efficiency in operations and maintenance, enhanced reliability of signaling systems, and improvements in overall safety. Furthermore, in upcoming years, Industry 5.0 will intensify human-machine collaboration, which is believed to lead to more human-friendly working environments. For railway transportation, which is the focus of this paper, the benefits of predictive maintenance and the shifting of repetitive/dangerous tasks to robots, will enhance sustainability and resilience [3].

However, digitalization efforts come with increased security issues as more digital services, communication protocols, and interfaces become involved. For instance, as discussed in [4], the Railway Safety Transport Application (RaSTA) protocol provides, according to the authors, limited security features. Additionally, security breaches can lead to more severe consequences in comparison with classical IT systems as compromised safety mechanisms in railroads may cause loss of life and large financial damage. As an example, a recent attack against the Global System for Mobile Communications-Railway (GSM-R)-based communication link [5] led to large-scale shortages in railway traffic in northern Germany, which subsequently caused financial loss to the operator company. Moreover, in future railway systems that will interconnect sensors, controllers, and AI, cyber security incidents will likely become even more pronounced. Firstly, due to interconnections to the physical world, cyber-physical attacks might give an attacker a way to cause physical damage to infrastructure or even put human life in danger. Secondly, the risk of data breaches increases as sensitive data are collected, communicated, and stored in large volumes. Access to such data, e.g., proprietary data, might lead to leaks of confidential information.

To avoid security breaches, the EU has issued a network and information security directive [6], which aims for higher protection levels at critical infrastructures, such as railway networks. In addition, technical specifications on handling railway cyber security were prepared by the Cenelec TC9X working group [7]. In recent years, research efforts have been devoted to improving confidentiality, integrity, and the availability of railway communication systems. In [8], an active intrusion detection mechanism was proposed, which relies on a security layer built around the critical functional blocks with an additional secure communication channel. While such a mechanism can efficiently detect attacks, its implementation requires further communication bandwidth and equipment, which also needs to be secured. In a similar fashion, [9] demonstrated a framework that implements a security overlay for safety-critical embedded components. Moreover, lightweight encryption [10] and authentication algorithms [11] provide end-to-end encrypted communication in railway networks. Finally, mitigation techniques against denial-of-service (DoS) attacks have also been discussed, e.g., in [12], where the authors established a three-level filtering technique to restore Railway Signal Safety Communication (RSSP-II) protocol communication under a distributed DoS attack.

Due to the critical nature of the service and the potential level of impact of the damage caused, it is necessary to think about train transport in a comprehensive way and to ensure the maximum possible state of safety using a combination of different tools [13]. Individual attacks are linked, among other things, to the rise of services mediated through the Internet of Things (IoT) or Industrial IoT (IIoT) [14]. In recent years, research has been focused on intrusion detection system(s) (IDS) using machine learning and neural networks among others [15,16]. Our goal is to create a tool capable of performing individual data processing from sniffing traffic and identifying individual anomalies into specified categories. The claim of the tool is mainly to limit the delay caused by the classification of the data traffic. For this reason, the data are obtained by sniffing the data traffic and potential countermeasures can be triggered depending on the number and severity of the identified anomaly.

Although there are many novel state-of-the-art security-enhancing features and frameworks, one of the research gaps in railway security, as identified by the recent review paper on the subject [17], is the lack of experimental validation. While some works include experimental data, they are often extracted out of a software simulation, whose behavior might deviate from real-world scenarios. Furthermore, to the best of our knowledge, only a fraction of cyber security-related works have considered dedicated security features for axle counting systems, which are important components regarding railroad safety. With the above points in mind, in this paper, we set out to improve the state-of-the-art as follows:How to classify and handle targeted attacks?How to avoid overfitting the model and adapting it for real use on data that have not been trained (in terms of time values)?How to use the axle counting system to detect and classify security incidents in this network?How to perform a thorough evaluation of the model and the effect of overfitting on the model?

The rest of the paper is organized as follows. In Section 2, we review the state-of-the-art methods regarding anomaly detection in railway systems. Section 3 presents the methodology, which is followed by the presentation of the experimental results in Section 4. Finally, our conclusions are presented in Section 6.

## 2. ML-Based Intrusion Detection Methods in Railway Environment

This section presents a review of the literature on the detection of anomalies in the railway environment. In this review, we limit the scope so that only those works that are closely related to our focus are included. As a result, we did not consider issues related to train delays or physical anomalies that were detected from image data.

Our work focuses primarily on anomaly detection in the area of cybersecurity in railways. This area of anomalies has been the focus of several works in recent years [18,19,20,21].

Gómez et al. [18] created their own physical test environment by simulating an electric traction substation for this purpose. Using this test environment, they generated datasets containing S7comm and Modbus TCP network traffic data. To introduce anomalies, they performed several cyber attacks in the areas of reconnaissance attacks, false data injections, and replay attacks. For anomaly detection, they chose two machine learning approaches, namely supervised learning and semi-supervised learning. In their work, they compared ML algorithms, such as random forest (RF), support vector machine (SVM), one-class support vector machine (OCSVM), and isolation forest (IF), as well as one deep learning model, i.e., the dense neural network (DNN).

Heinrich et al. [19] focused on an anomaly detection system to defend against semantic attacks in railway signaling networks. The data they processed stemmed from a physical test environment that was close to a real solution. The authors of this paper processed network traffic data from a signal box. The dataset itself is not provided in the paper. In their work, they adopted an anomaly detection approach based on a supervised deep learning technique based on artificial neural networks (ANNs).

Islam et al. [20] primarily focused on the potential of anomaly detection cyber-attacks on the Internet of Things (IoT) in a railway environment. They presented two datasets, one focusing purely on IoT and the other on IoT in railway environments. Unfortunately, the second dataset mentioned is not referenced. The data they process were also network traffic data. For anomaly detection, they used unsupervised learning approaches based on an extended neural network (ENN), convolutional neural networks (CNNs), long short-term memory (LSTM), and DNNs.

Jiang et al. [21] used the Electra dataset for their work, which was presented by Gómez et al. [18]. In their work, they focused on a supervised learning approach based on SVM, OCSVM, RF, IF, NN, and generative adversarial network (GAN)+DNN methods. They mainly focused on the use of a denoising autoencoder (DAE), a synthetic minority oversampling technique (SMOTE), Tomek link (T-Link), and extreme gradient boosting (XGBoost) mechanisms to improve the prediction of individual models.

None of the previous works mentioned testing and verification of their solutions in real operations, which is a key point for our work.

The other mentioned works did not focus on anomaly detection within cyber security, they mainly focused on anomalies in functionalities and health of the whole system, the system functions themselves and, in general, on fault detection in the railway industry. These works also focused on different types of data than network traffic.

Da Silva Ferreira et al. [22] dealt with a comparative analysis of unsupervised learning methods for anomaly detection in railway systems. Within this approach, they compared K-means, a self-organizing map, and autoencoder algorithms. They focused on operational data (electrical samples) from track switches. The data they used were from real traffic. The dataset in the paper is not provided publicly. The paper did not mention whether the methods were tested in real traffic. Oliveira et al. [23] discussed the use of unsupervised techniques to detect faults. They focused on isolated forest and autoencoder algorithms. The data they used came from real traffic in Brazil. They focused on operational data from areas such as the wheel impact and load detector (WILD), hot box and wheel (HBW), and the rail BAM acoustic bearing detector (RB). The dataset was not provided, and they did not mention whether they tested their solution in a real operation. Appoh et al. [24] discussed the state of system health and the prediction of potential consequences based on the condition of the wheel flange lubricator. Their focus was on supervised methods, namely the SVR algorithm. They used numerical data such as weight and distance obtained from real traffic in the United Kingdom for processing. The dataset was not provided, and they did not mention whether they tested their model in real traffic. De Santis et al. [25] worked with real data from Italian railways, where they focused on data in the form of pressure sensor signals. They focused on supervised learning, namely on the SVM algorithm. They did not provide a dataset in their work and did not mention testing their own model in real traffic. Bel-Hadj et al. [26] focused on anomaly detection based on signal data from strain sensors on a railway bridge. They dealt with unsupervised sparse convolutional autoencoder (SAE) techniques and Mahalanobis distance (MD) techniques. The data they worked with were collected from real traffic. In this paper, they did not provide a dataset and did not mention testing their model in real traffic.

A summary of these papers can be seen in Table 1. The table is divided into two parts. The first part focuses on approaches dedicated to anomaly detection of cybersecurity incidents. The second part focuses on anomaly detection in the context of system health, system functions, and fault detection. In the table, the author’s name and the reference of the paper are given as identifiers, followed by the year of the publication of the paper. The ‘ML Focus’ column describes which machine learning method was chosen followed by a column that describes the algorithms used. The ‘Area of Interest’ column describes which elements of the railway the paper focuses on. In the ‘Type of Data’ column, we define three data options: network communication data (data from network communication protocols), operational data (numerical data from sensors in integers or floating point format), and signals (measured data based on sensed physical quantities and direct signal processing for algorithms). We describe the origin of the data, where the question is whether the data were collected from a testbed or real traffic and whether the dataset was available to the public. In most cases, datasets were not made public. The last column describes whether the resulting model was tested in real traffic and not just tested on the test dataset. This information is important for our work because of the objectivity of the results.

## 3. Methods

### 3.1. Axle Counting Infrastructure

To achieve safe railway traffic, it is necessary for the different train movements to be separated from each other. There are three different theoretical principles of train separation—separation by the relative braking distance, separation by the absolute braking distance, and separation into fixed block distances. Out of the three principles, the separation into fixed block sections is the most common method in Europe [27]. For this purpose, a railway line is divided into several block sections, whereby only one train is allowed to be in a section at the same time. The different sections are separated from each other with signals, which act as indicators for a train driver, i.e., whether a train is allowed to enter a section or not. To ensure that a block section is clear, several forms of track clearance detection technology can be used [27]. The two most established methods for track clearance are track circuits and axle counters. Track circuits are based on the fact that the rail and the axle of a train are made out of steel. An electrical source is used to send electrical currents over the tracks to show that, at a picked relay, the section is free of any train movement. If a train axle is in the section, it produces a shortcut and the relay is dropped, so the train can be detected. The second technique for train detection involves axle counters, which consist of two inductive sensors. These double sensors not only increase the reliability, but they can also detect the direction of a train. Therefore, at the beginning and end of every section, there has to be an axle counter. If a train enters the section, every axle of the train is counted at the axle counter, and the section changes to the state occupied. If the train leaves the section, the axle counter counts every axle again, if the difference of both sums is zero, then the state of the section changes to ‘clear’. To ensure these functionalities, every axle counter needs a digitizer, which is located in the direct surrounding, which digitizes the analog signal. An electronic evaluator, which is connected to the two digitizers of the section, uses the information and calculates if a track section is occupied or clear [28]. To explain the functionality, a practical example of the use of axle counters is shown in Figure 1. The location of the axle counters in a realistic Scenario Is illustrated in the upper part of the figure. The lower part is depicted with a gray background and, therefore, shows an example illustration of a station dispatcher. On the second track, there is a train that is leaving the station on the right side. At this moment, the train occupies the section of the second track and the turnout area. Therefore, for the station dispatcher, it is not possible to send another train on the first or second tracks. A train from the left side can follow the first train to the second track only when the first train leaves the axle counter section of the second track.

### 3.2. Vesper Man-in-the-Middle Detection Tool

Vesper is a readily available software tool used to determine anomalous changes in the LAN network structure [29], i.e., installation of passive traffic analysis components or devices, which perform active man-in-the-middle attacks. In this work, our intention was to utilize Vesper to augment the machine learning-based attack classification. The basic functionality of Vesper is based on an ICMP packet round-trip-time (RTT) analysis, which captures latency and jitter induced by network devices. Deviations from the normal latency characteristics (due to, e.g., the data processing delay of an attacker) will cause anomalies in the recorded RTT measurements, which will ultimately reveal the presence of an attacker.

The timing analysis in Vesper draws similarities to reverberation modeling of acoustic environments, where a system model can be extracted by analyzing an impulse response with digital signal processing methods. In LAN networks, data buffering, data caching, and data processing cause delays and jitters to network traffic and, hence, a similar timing model can be extracted using, e.g., ICMP echo request (impulse) and ICMP echo response (impulse response) packets. Additionally, to arrive at richer timing statistics, the echo packet size is modulated based on maximum length sequences. According to the experimental results of the authors of Vesper [30], the different packet lengths lead to different RTT distributions, which are useful later during the anomaly detection phase. The anomaly detection process of Vesper is composed of four main parts, which are briefly described below.

System orchestrator: During the initial step, a random IP address of a network host is selected to be analyzed.Link prober: The link prober establishes ICMP traffic between the Vesper node and the network host. Several ICMP echo probes are sent and the echo response packets are recorded with Linux operating system timestamps with nanosecond accuracy.Feature extractor: The timing data from ICMP packets are extracted and the following features are evaluated: impulse response energy, mean RTT for the largest ICMP packets, and log-likelihood of the jitter distribution.Host profiler: An autoencoder artificial neural network reconstructs the original data distribution from the recorded features.  

The actual steps to establish anomaly detection in a LAN network with Vesper involve creating a population in the form of a list with network host IP addresses. Afterward, during the training phase, the Vesper echo analysis is performed to build host profiles, which are stored individually for each host. Finally, continuous network anomaly detection is performed by following the steps described above.

### 3.3. Traffic Classification

Based on the knowledge of the behavior of the targeted system, we determined the classes into which traffic will be classified using neural networks and machine learning techniques. From a formal point of view, two basic classes can be defined, corresponding to normal traffic (without the presence of an attacker) and traffic influenced by an attacker (the emergence of malicious behavior). Furthermore, normal traffic can be divided into “clean traffic” and “section occupied”. The term “clean traffic” will be further used to refer to traffic where only status data are sent within the system without the presence of a train moving between sections (in other words, clear track without the presence of an attacker). The term “section occupied” will further refer to a condition where there is movement of a train between sections (axle counting system detects movement) without the presence of an attacker. The second class, the traffic affected by the attacker, is further divided into four main categories in this paper. The first category refers to DoS attacks on individual system elements, the second category refers to replay attacks in which traffic with a passing train or free track sections is replayed. The next category includes man-in-the-middle (MitM) attacks and the last category includes network scans. The target network classes into which network traffic is divided in this paper can be referred to as:Without the presence of an attacker:-Label 0—clean traffic.-Label 1—section occupied.With the presence of an attacker—the emergence of malicious behavior:-Label 2—DoS attack,-Label 3—Replay attack,-Label 4—MitM attack,-Label 5—Network scanning.

In the case of the DoS attack, we used the hping3 tool to simulate various flooding attacks targeted at the units of the axle counting system. The replay attack used the tcpreplay tool to replay the already captured data. During this attack, the clean traffic or section-occupied traffic was replayed. A combined attack was also included in this category, which used a MitM attack to block messages from individual members to the parent/control station while launching a replay attack. This attack was used to achieve a “simulation” of a moving train within each section (MitM + replay of the section-occupied traffic) and also a “simulation” of a clear track (MitM + replay of the clear traffic). MitM attacks use the ettercap tool, where the created filter was used to remove network traffic originating from the selected nodes of the network. The last label covers network scanning performed by the nmap tool.

### 3.4. Selected Approach

Several techniques are used to classify network traffic [31]. In this paper, we present a traffic classification capability where detection is based only on the data processing of a safe railway transmission protocol used in our testbed [32]. Thus, it is a packet-based processing technique. This technique allows the detection of even minimal changes in a packet and does not dilute the induced effects among other packets. This kind of processing can be critical, for example, to detect only one packet that is sent into the network by an attacker. For packet classification, we focused on machine learning (ML) methods and deep neural network (DNN, NN) methods.

To do that, we needed to first classify which data we were capable of processing from the point of view of the protocol. Table 2 shows the data, which were taken from the captured data generated on our physical testbed. Data were taken from the physical layer (L0), network layer (L3), transport layer (L4), and application layer (L7).

### 3.5. Attacker Model—Attack Vectors

Based on the knowledge of the testbed, which we used to generate the data, we determined the attack vectors that the attacker could use. We separated our testbed in total into three layers—the physical layer, the aggregation layer, and the processing layer; see Figure 2.

The main objective of the physical layer is to detect the individual axles of the train using physical principles, such as the use of capacity changes. Thus, it is the most basic and pivotal part of the whole system. The purpose of the aggregation layer is to process individual signals transmitted from the physical layer and then convert them from voltage values (serial communication) to digital values using the appropriate transmission protocols. The aggregation layer obtains data from the processing layer, ensures their visualization, and supervises the whole system. The data flow can be as follows, the first axle passes through the axle counting system, the data are processed by the parent units (aggregation layer) and then processed by the processing layer, which detects the passing axle, increments the axle counter, performs a section closure check (signal marked red) and finally informs the station dispatcher. The attacker can potentially target every part of the model. From a physical point of view, the attacker could physically damage any part of the system. Devices in the physical layer are especially vulnerable to physical attacks. If the attacker will not exploit physical damage to the device, a possible target is the physical connection of an attacker-controlled device to the aggregation or process layer. In these two layers, the attacker might act in two modes, passively sniffing network traffic or actively handling network traffic. In our selected scenarios, we focus mainly on active attacks, where the attacker uses DoS attacks, replays captured communication, establishes MitM, and performs network scanning. We named one of our attacks the silence attack. This is an attack that uses a MitM attack and actively blocks communication (using an ettercap filter) originating from the aggregation layer, the process layer loses connectivity and the operators of these systems are unable to use the axle counting system for the duration of the attack. A more advanced attack is then a combination of a silence attack with a replay attack, where the selected attacks are intentionally cast or released. Thus, this is the attack with the greatest potential impact and is also very difficult to detect (using conventional methods). The objectives can be identified as follows: physical damage to the equipment, disabled infrastructure, disabled service, using a fake service (empty, section occupied), and a combination of them.

## 4. Experimental Results

### 4.1. Testbed

For testing our attack detection methods, a testbed with real-world axle counting components was built. As visible from Figure 2, the selected constellation of six-axle counting sensors represents the use-case introduced in Section 3.1. The wheel detection system includes the following components: evaluation boards, communication boards, an interlocking simulator, and a diagnostics system. The physical axle counter sensors are replaced in the testbed by fixed resistors, which allow for the simulation of voltage signals induced by train wheels. The simulated physical sensor input signals are digitized and processed in the evaluation boards. The axle counting data are further communicated to the interlocking simulator and diagnostics via communication units. Between the interlocking simulator and communication units, the data are carried with a safe railway protocol, and further, toward diagnostics, the data are carried with a proprietary UDP-based protocol. With the diagnostics system and the interlocking simulator, a user is able to monitor the status of the axle counting sensors as well as the status of the track sections in real time.

### 4.2. Data-Preprocessing

To be able to classify the network traffic into determined classes, we needed to generate an appropriate dataset consisting of data from each class. To capture all the data, we used the network tap on the main switch; a PC was connected to this port, which was only used to capture network communication. Based on the targeted classes, we created in total 6 scenarios (setup), based on the attack or type of traffic we needed to capture. After collecting network capture data, we used the tshark tool to transfer selected data from the pcap file to the csv file. Subsequently, the data-preprocessing process was carried out, where the selection of the appropriate data-preprocessing procedure (in particular, the selection of key features) was carried out in several iterations. The preprocessing block itself consisted of the selection of appropriate features, parameter processing (operations on individual features), and CSV cleaning (operations of removing incomplete and inappropriate data, and replacing empty values with a placeholder value). The selected features were mentioned above in Table 2. The last operation of the data preprocessing was labeling based on the class being processed. After each class was processed, the individual classes were merged into a single dataset and then a positional mapping (randomly changing the sequence of rows) was performed to obtain the final dataset. The entire process and its main blocks are shown in Figure 3.

### 4.3. Key Parameter

Among the basic requirements of the final solution was to create a machine learning and neural network model that would be as usable as possible in a real environment (at least a testbed). As revealed by experimental testing, the use of timestamps significantly increases the accuracy of the model (values very close to 99%), but in real use, these models achieve only low accuracy and are, thus, not suitable for real use (the real values do not exceed 50%). The use of timestamps (as one of the inputs to the model) is thus very risky in terms of creating too much dependence on the model on these values. Even when using relative values (Δ values—the difference in the values of the time instants between the reception of each packet), the results taken by testing the trained model on the testbed (no data during the training, testing, or validation were used) were not sufficient. For that reason, the feature epoch time (mentioned in Table 2) had to be removed. Subsequently, using statistical tools, it was found that the RX and TX Timestamp values (features) alone did not provide any information and thus they were not suitable candidates as input parameters for the resulting model. However, the combination of these parameters is crucial for traffic classification. This is due to the different timing effects on these values by the attacker. Moreover, these impacts are slightly different for each attack, and, thus, the attacks can be separated using additional parameters. Furthermore, the attacker is unable to reduce the impact on these parameters in any way due to the impact on “normal traffic” by interfering with the communication. Figure 4 and Figure 5 show the dependency of the calculated differences of the *RX* and *TX* timestamps (y-axis) depending on the number of their occurrences in the input dataset (x-axis) for each traffic class. This new feature (marked as γ) is calculated as the difference between the *RX* and *TX* values in the absolute value obtained from the packet under consideration:(1)$$\gamma =abs\left(RX_{timestamp}-TX_{timestamp}\right)\left[\mathrm{ms}\right].$$

From both figures, it is visible that the presented method is able to distinguish individual labels based on the calculated value. Figure 5 differs from Figure 4 in that it reduces the number of visualized labels for greater clarity. Values of the RX timestamp, and TX timestamp declare the points in time at which the packet was processed within the receiver and sender system. The attacker in the network/infrastructure causes a delay in packet generating/processing, and this delay is not only detectable but can also be categorized into individual classes.

### 4.4. Deep Neural Network Approach

To provide packet classification, we used the DNN approach. We used Keras [33] software to build the DNN structure using the Python programming language. The whole approach is based on processing packet-by-packet, where the individual features are taken from the CSV dataset (when the model is being trained). This dataset consists of numerical values that are first normalized, and then the entire DNN structure is used to predict the packet being processed. To do that, we chose a multi-class classification with one-hot encoding as the output (label). This method ensures that the output class is always one of all possible classes. In total, there are six output classes into which the processed packet is classified, so the output might be “100,000” in the case when the packet is classified as clean traffic (only one class is set to the true value at a time).

Figure 6 compares the samples from the tests performed according to the input features used. In total, there are three types of experiments with a number and kind of input features. The chart is divided into three sections (violet, red, and green), with each section representing a different approach to data processing. Within the chart, the top six results for each approach based on experimental testing are plotted. The first was based on the use of a Δ value calculated as the difference between the moments of receipt of each message (marked violet) with 22 input features. With this type of input data, the models reach an accuracy of approximately 81% based on the testing data, but the evaluation conducted with the new testbed data demonstrated the model’s inability to make quality predictions based on unseen data during the training of the DNN model. The second approach (marked red) was done using only 21 input features, where the receive time was removed and values of RX and TX timestamps have been replaced by new values where the original values have been subtracted from the time of receipt. This approach reached similar results from the accuracy point of view but the models were not applicable in the real environment due to the low recognition ability of unseen data. The last approach (marked green) was built on the calculated γ value, see Equation (Equation 1). In this approach, only 20 input features were used (receive time, RX timestamp and TX timestamp features were replaced by the calculated γ value). Using this new feature, the accuracy decreased to 65.52% but the ability to classify traffic into determined classes was preserved. The highest accuracy values (the last two columns) were created by manually subtracting the minimum value from the input dataset from all records in the dataset (for the corresponding γ column). However, this approach has again caused a reduction in packet recognition capability within the dataset over unseen data (packets). Therefore, the best accuracy based on testing on unused data during model training and model testing was 69.52% using 20 features with the γ value.

Figure 7 shows a graphical visualization of the DNN structure. This structure is based on many experimental tests and is also built for the approach with the γ feature (mentioned above). So in this case, there are only 20 input features. The structure starts with the normalization part (layer). The goal of this layer is to scale the data into a distribution centered at 0 with a standard deviation equal to 1. This function is executed on the training data to create a normalization layer that will be used later. Usage of this layer ensures scaling (ranges) between all values without corrupting the data (distortion, breaking ties). Furthermore, the data pass in total through 6 hidden layers and two dropout layers. The aim of the hidden layers is to create a structure depending on the input data to perform a correct classification of the data. A hidden layer consists of neurons (in the figure marked with blue circles), in our case we used the fully connected DNN. The aim of the dropout layers is to avoid overfitting the model (makes the model unable to work/efficiently predict over unseen data). In our case, we used two dropout layers with the rate of dropout (setting inputs to 0 with specified frequency) equal to 0.3. The output layer has only 6 neurons, so the output matches the number of the output classes into which the data are being classified. The input layer and h2 use the “relu” activation function, and h1, h3, h4, and h5 use the “tanh” activation function.

For our purpose, we separated input data (in total) into three groups—training, testing, and validation data. The training data were used while the DNN model was created. Furthermore, the validation data were used while the structure of the DNN was tuned, and finally, the testing data were used while the created model was evaluated (these were not used during the model creation process). In our approach, we also created a custom callback, so we evaluated the testing data at the end of each epoch. Using this approach, we can evaluate the model during the testing and also save the best-trained model. The results achieved are shown in Table 3. As mentioned above, we classified the traffic into 6 classes, but we also implemented a DNN model with 5 and 2 output labels. In the second scenario, the first two labels were merged. Thus, normal traffic is recognized (clean traffic and section occupied), and then there are the same attack classes as in the first scenario. In the last scenario, there are only 2 classes for regular traffic and attack. As the number of classes decreases, the accuracy of the model increases. This is mainly due to the difference between the first and the second scenario; in the first case, the second class (section occupied) was incorrectly labeled as the first class (clean traffic).

The calculations in the table were obtained from the confusion matrix using the one-vs-rest method. Individual metrics for each class were obtained separately, which is the main contribution of the ability to correctly classify each class independently. Thus, based on the matrix, the overall accuracy can also be obtained as the sum of the correctly labeled classes divided by the sum of all elements of the test set. The overall accuracy in Scenario I was equal to 69.52%, Scenario II increased the overall accuracy to 85.11%, and Scenario III also increased the overall accuracy to 92.02%. There was also a need to change the number of input features in Scenario III, due to the highly imbalanced input dataset.

If the dataset from the first and second scenarios were used, there would be a significant imbalance in the dataset. thus training a model incapable of proper classification into classes. This imbalance of the dataset in the model creation phase causes the model to be unbalanced in terms of its subsequent implementation in the “real” network. The imbalance of the dataset is caused by the uneven representation of the classes in the dataset. If the same dataset was used (as for scenarios I and II), class I (regular traffic) would be represented by 12,088 samples and class II (attack) by 95,048 samples. Thus, legitimate traffic would account for only 11.28% of the total dataset samples. The goal is always to create a dataset that is as diverse as possible, representing all recognized/identified classes equally. After modifying the dataset, the number of legitimate traffic (regular traffic) was changed to 92,428 samples and the attack class contained 84,029 samples. This resulted in the occurrence of legitimate traffic in 52.38% of the newly modified dataset.

According to our experiments with DNN-based anomaly detection, the classification of the man-in-the-middle attacks in our use-case is challenging as the network traffic features show only minimal deviations. Therefore, in order to optimize the detection accuracy further we propose a combined detection approach, which substitutes the passive DNN-based attack classification by the active man-in-the-middle detection implemented with the Vesper tool introduced in Section 3.2. With the pre-trained host network delay profiles, an accuracy of 100% for the man-in-the-middle classification could be achieved with active probing, which is in accordance with the published results in [30]. As shown in Table 3, increases of 10.89% and 7.88% in classification accuracy were achieved. In Scenario I, the overall accuracy increased from 69.52% to 77.41%. In Scenario II, the overall accuracy increased from 85.11% to 92.29%.

### 4.5. Machine Learning Approach

Within the classical machine learning algorithms approach, the dataset handling procedure was the same as for the DNN in Section 4.4. The approach differed only in the third phase, where models based on classical machine learning algorithms had very poor results or were overtrained; thus, the results were not applicable in real operations. To provide packet classification using machine learning, several classical algorithms have been compared, namely: support vector classification (SVC), k-nearest neighbor (kNN), random forest (RF), AdaBoost classifier, decision tree classifier (DTC), Gaussian Naive Bayes (Gaussian NB), Bernoulli NB, and MULTI-LAYER PERCEPTron (MLP). For processing, we used the scikit-learn library for the Python programming language. Considering the set of possible behaviors on the railway as described in Section 3.3, we chose a multiclass classification approach, where the different types of traffic were labeled with numbers 0 to 5 (that is, 6 labels). The data set was always divided into three parts for all phases: 70% training data, 15% validation data, and 15% test data. The description of our experimental part with the machine learning approach is described in three parts, as it was for DNN. These phases depend on the features used in the datasets, and thus specific features were used for each phase. It is important to mention these individual phases here in the general context of the results in real-operation testing.

In the first phase, all of the features (22) within the data set were retained as shown in Table 2. When evaluated in the validation and test datasets, we obtained accuracy_score = 1.0 for the random forest algorithms and accuracy_score = 1.0 for the kNN algorithms. This excessive accuracy was obviously misleading, so we tested the models in real operations. In this test, none of the algorithms was able to mark the traffic correctly. Both algorithms labeled different traffic all of the time as one label (replay attack). This showed that the algorithms were overtrained. The overtraining was due to the use of time information, which made it possible to predict within the test data set very accurately but made the trained model unusable in real operations. We then removed all time information from the dataset, but as might be expected the accuracy_score for the best models was 0.59 for kNN and 0.57 for Random Forest. However, in a real operation, the models were again unable to identify the different types of traffic and labeled all traffic as one single label. Therefore, some ’time’ information had to be retained. The second phase was the most successful in terms of real-life use of classical machine learning algorithms, namely the kNN algorithm, which achieved the best results. At this phase, time information within the data set was removed, and new features with time differences were created. Epoch time was removed and the values RX and TX timestamp have been replaced by new values where the original values have been subtracted from the epoch time. A total of 21 features were used as input. During this phase, a set of different algorithms was tested. According to the accuracy_score evaluation, the best algorithms were RF and DTC, which again had a value of 1.0. However, when tested in real operations, these algorithms acted in the same way as in the first phase and were not able to recognize different types of traffic. Everything was marked with the same label. The best-performing algorithm in real operation was kNN, which had an accuracy_score of 0.75. When labeling in real operation, this algorithm was able to label all types of traffic and attacks. Errors occurred during the marking but were not significant, and the model was able to react to changes in traffic. During parameter tuning of kNN, overtraining occurred and the model was not able to mark different types of traffic. The accuracy_score of this model was 0.99. The best parameter settings were as follows: metric: Manhattan, n_neighbors: 9, weights: distance. However, in a real operation, the results of this model were not good (it was overtrained). As already mentioned, much better results were achieved by the basic kNN model. Table 4 shows the most successful models for this phase, but as already mentioned, RF and DTC have the best results but are not applicable in real operations. The RF result has been added to the table for comparison only.

The third phase was based on a feature with the γ value within the dataset. There were 20 features as input. Compared to DNN, the classical ML models had very poor results. The RF and DTC algorithms were again overtrained and not applicable in real operations. For kNN, the resulting accuracy_score was very low at 0.54. When attempting to tune the parameters, it was possible to obtaining to 0.99 when again, as in the previous phase, overtraining occurred and the model was not usable in real operations. In Table 5, we present a comparison of the results from the second and third phases for each algorithm. In this table, we provide a summary of all of the classification methods we compared and their results.

### 4.6. Tool for the Implementation of the Trained Model

To provide a legitimate evaluation of the trained model, we developed a tool that uses the trained model for packet classification. This tool is built on a pyshark tool that sniffs the network interface that is connected to the network tap. The safe railway protocol-related data are preprocessed and then the data go as input into the trained model. Then the model provides a classification of the packet under consideration. The data flow is visualized in Figure 8. Based on the experimental testing of ca. 4000 packets, the average processing delay is ca 2.0 ms, the median value is equal to 1.7 ms. This time delay represents the delay induced by the system to classify the data packet. This time is measured as the difference between the time at which the packet is classified (labeled) and the time at which the packet is received for processing. Due to the use of devices in a virtual environment, there was an additional delay caused by individual packets waiting in the queue before being processed by the system. This delay was up to 1s within a single packet considered. However, to achieve more efficient processing, the use of more powerful equipment, the use of parallel processes, and the optimization of the network tap to send only selected packets seems to be a suitable solution. Further optimization is planned in future work.

## 5. Discussion

From the results of our work, we can see that if we want to use classical ML algorithms, it is preferable to choose the same approach as in the second phase of our testing, where the best model that is able to recognize traffic even in real operation is the model based on the kNN algorithm whose accuracy_score is 0.75. The DNN model was only applicable in real operation in the third phase of the testing, which was based on the value γ, where the model achieved an accuracy_score equal to 0.695. For the first two phases, the model was unable to correctly detect different types of traffic in real operations. In the analysis of the results data, we found that in the third phase, the DNN model loses the most in terms of recognizing two labels from the category “without the presence of the attacker”, namely: clean traffic and section occupied. With the ML approach, the misinterpretation of the data was not as clear and the errors were spread across different labels. Therefore, we decided to experiment with different numbers of labels within DNN and created three scenarios. Scenario I was a common scenario with the labels. In Scenario II, we decided to merge the clean traffic and section occupied labels into one label and create a new label called regular traffic since it is the traffic without an attacker present. In this scenario, the model achieved an accuracy_score of 0.85. In Scenario III, only two labels were used, namely regular traffic and attack in which case the model was the most successful with the accuracy_score =  0.92. From this, we can say that our solution is quite successful in detecting whether it is a cyberattack or standard traffic. Our model is not as accurate for identifying the different types of attacks and, therefore, achieves a maximum accuracy of 0.85. However, we can say that it works quite well in real operations.

The time information had the greatest impact on the evaluation of the results, and it was critical for both approaches to choose an appropriate time information representation. For the kNN model, which was the most suitable of the classical ML algorithms, it was preferable to choose the approach described in the second phase. For the DNN approach, the value γ had to be calculated to train the model to be able to recognize different types of traffic in real operations. Time parameters influenced the results the most. In this paper, we also tested the approach without any time parameters in phase one, but the models performed very poorly in both approaches. It can also be seen from our results that training the models and then testing only on the test dataset is insufficient. For classical ML methods (e.g., RF or DTC), we could claim that the results are very accurate (the accuracy_score of both models is 1.00), but these values are misleading and in real operation it turned out that the models are not even able to detect different operations, they marked everything as one kind of traffic. It is, therefore, necessary to use the correct time features when processing the data to avoid overtraining. In the context of anomaly detection in test environments, the model may be overtrained based on certain features. If the model is not tested in real operation, this error may not be detected by the model’s author. Therefore, it is not reliable to test the model only on the test and validation dataset. If there is no other option, a lot of attention must be focused on creating a very high-quality test dataset where the data do not overlap with the validation and training dataset.

The creation of a suitable dataset represents the most critical point for creating a proper model capable of applying the created structures to unused data. The dataset for model building requires the occurrence of samples from each classified class in numbers as balanced as possible. Otherwise, there is a risk of overfitting the model. This dataset also needs to contain as diverse data as possible in terms of individual states, i.e., it is advisable to include in the dataset all states that may be set within a classified class (to capture the full range of potential states). Subsequently, the dataset must be cleaned of possible “anomalies” in terms of data (these may be critical states, outlier situations, etc.) and further normalization of the values is performed in order to stabilize the gradient descent step and to ensure faster convergence of the model. The dataset cleaning operation is the second most critical operation in terms of dataset creation, after the collection of suitable data. It is used to remove data anomalies that could introduce some degree of error into the model. For the purpose of model creation and development, the training and validation sets are used and the test set is used for the final evaluation of the quality/success of the model. For the sake of quality testing of the decision/classification capabilities of the trained model, the test data must not be included in any way in the model-building process, otherwise, there would be a risk of overestimating the model due to the knowledge of specific values instead of the structures and constraints created. For this reason, we also performed testing on the workplace itself instead of using only the recorded data.

The papers most similar to our approach are those included in Table 1 in the cybersecurity section [18,19,20,21]. In these works, the authors used data from network communication protocols as data sources. In our work, we also targeted network traffic of axle counting data, which are not included in the mentioned works. Hence, our focus on the axle counting part of railway sensors distinguishes us from the mentioned works. As a test environment, we had a physical test environment simulating axle counting. In papers [18,20,21], the authors, such as us, dealt with supervised learning ML methods. However, neither paper focuses on the kNN algorithm, which in our case came out best of the classical machine learning algorithms. In our work, we consider it generally beneficial to test models in real operations and thus be able to discard algorithms such as RF or DTC that were not applicable in real operations. This real operation testing had an impact on how to think about the data itself and especially the effect of time parameters on the results. We think that choosing the right approach to time parameters is critical to the results in real operations.

Finally, we recognize the value of our results in improving the resilience of axle counting systems. With an ability to distinguish between normal network traffic conditions and several attack types, the proposed intrusion detection system can potentially reduce the response time of an IT security analyst as fewer manual efforts are required. Thus, our proposed detection mechanism for multiple attack types is a suitable fit for, e.g., cyber defense centers dedicated to railroad applications.

## 6. Conclusions

Railway transportation, similar to other transportation systems, uses cyber defense mechanisms to increase security. These mechanisms provide additional security or provide more relevant information about the controlled zones. However, in the hands of an attacker, they can be a powerful weapon not only to disrupt security but also to induce life-threatening conditions. For these purposes, two methods (using machine learning and a deep neural network) were used for the classification of the packets on the network (supervised learning).

Based on the stated research questions, we concentrated on targeted attacks against axle counting systems (in total six classes, five of which represent the attacker, respectively, and the impact of individual actions of the attacker)—question I. We created a feature/parameter to increase neural network metrics, in this paper marked as the γ parameter—question II. We demonstrated successful attack detection in a real-world axle counting use-case implemented using off-the-shelf components—question III. We rigorously validated the intrusion detection performance with measurements based on real-world data and in a real-time operation—question IV.

Our presented approach is capable of classifying the network traffic in total into six classes with a total accuracy of 85.11%. This classification provides the possibility to distinguish between four attack-related classes and two legitimate traffic-related classes. One of the contributions of this work is the ability to perform data classification depending on the safe railway protocol, i.e., to classify the impact of attacks performed by an attacker using a protocol that was not the target of the attack itself. We also conducted many experimental tests focused on the appropriate selection of features, as well as their preprocessing. Subsequently, the input data and the model were validated on different data within the testbed.

In this work, we also presented an approach in which we performed packet classification using the γ parameter obtained from the traffic in conjunction with the safe railway protocol (the highest quality model capable of performing classification even on different data within the testbed). This parameter takes into account the various impacts on the safe railway protocol caused by an attacker and allows traffic classification into specified classes. Thus, in this case, the intrusion detection system and intrusion prevention system techniques that use thresholds would not be able to perform legitimate classification. Another contribution is the validation of the presented approaches based on the data from the testbed environment, so these data were not used in any phase (training, testing, validation). Moreover, our approaches do not use time-based parameters; the γ parameter was based on the delay between individual nodes that the attack caused. We also focused on classification with minimal delay; in our approach, we achieved a delay of 2.0 ms in terms of data classification.

The overall contribution of the paper might have some potential limitations that need to be considered when interpreting the results. The most important part of the work is the dataset used to create the machine learning and neural network models. The dataset was created based on the available operations within the experimental site, and as such, it may not be fully representative of all possible scenarios and states within the railway network. Our created dataset fully reflects our experimental environment in terms of individual network delays, devices used, etc. Therefore, the dataset used may differ from other sites. Therefore, when transferring the model created by us, it might be necessary to retrain the model to achieve identical results as in our case. Another potential limitation is related to the scope of the study. In our case, we focused on selected safety incidents and their classification in the railway network. Thus, the results and findings may not be fully applicable to all possible security scenarios within the railway network. Thus, the results and findings presented in this paper may not be fully applicable to all possible security scenarios within the railway network. In summary, while our study makes a significant contribution to the field of railway network security, several potential limitations need to be considered when interpreting the results. Thus, the created dataset may not be fully representative in terms of individual states and elements within the railway network, the focus of individual scenarios may not fully cover the whole range of possible safety incidents, and the change of the network may require retraining or adjusting of the model.

As part of future work, several additional steps for improvements and follow-up testing are being considered, such as further reducing the delay caused by classification and by waiting in the processing queue. Testing unsupervised learning methods is also being considered. We also plan to perform long-term testing (weeks/months) to create a dataset for further testing and refinement of the presented methods. Finally, integration of our solution within cyber defense centers is also being planned.

## Figures and Tables

**Figure 1 sensors-23-03122-f001:**
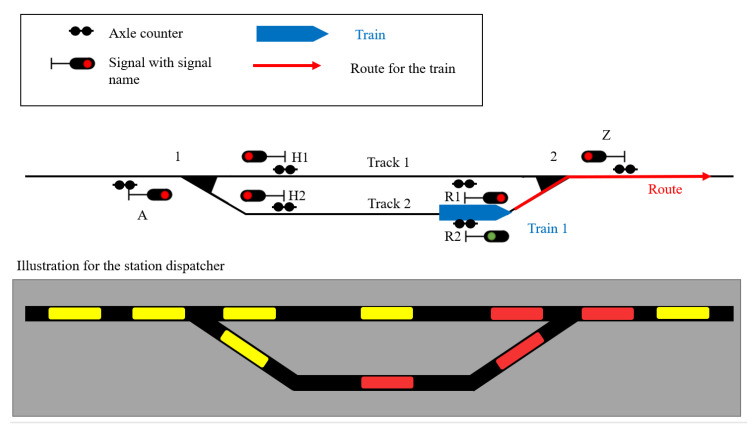
Practical use of axle counters in a railway station.

**Figure 2 sensors-23-03122-f002:**
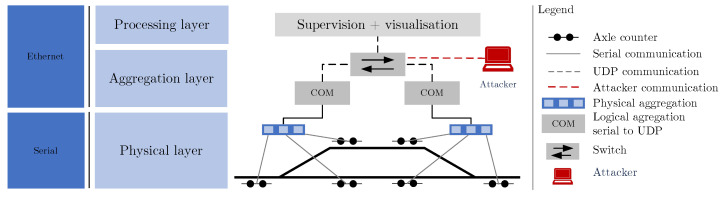
System diagram of the testbed and attack vectors that our proposed method covers.

**Figure 3 sensors-23-03122-f003:**
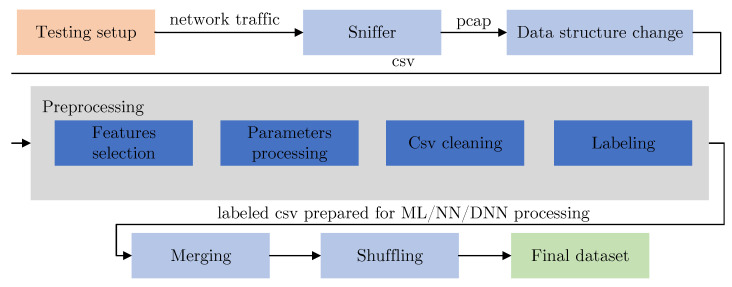
Data-preprocessing flow to create the final dataset.

**Figure 4 sensors-23-03122-f004:**
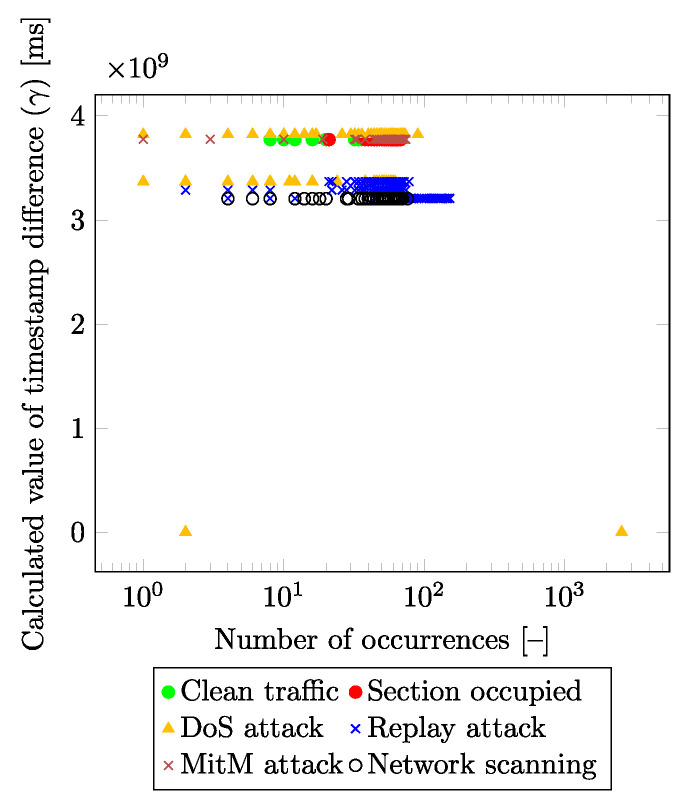
Distribution of timestamp difference values in the dataset (in terms of all labels).

**Figure 5 sensors-23-03122-f005:**
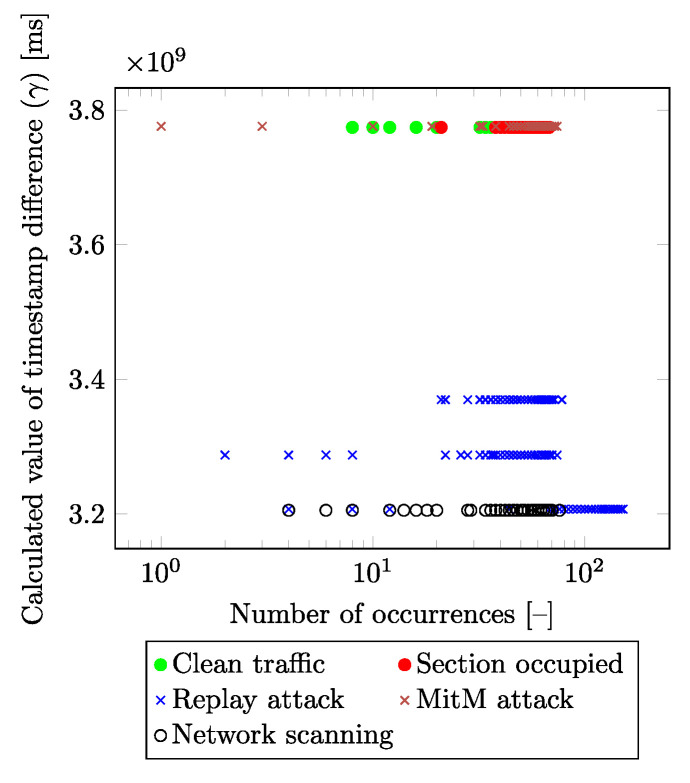
Distribution of timestamp difference values in the dataset (without DoS attack).

**Figure 6 sensors-23-03122-f006:**
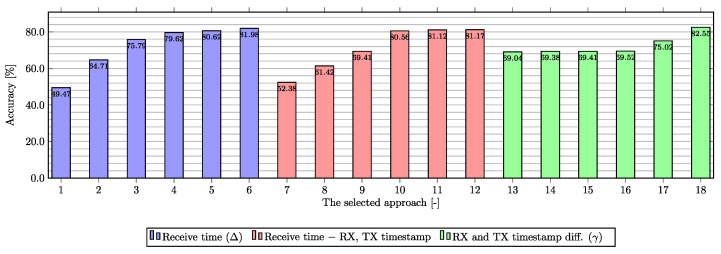
Comparison of achieved results using DNN approaches based on the testing data.

**Figure 7 sensors-23-03122-f007:**
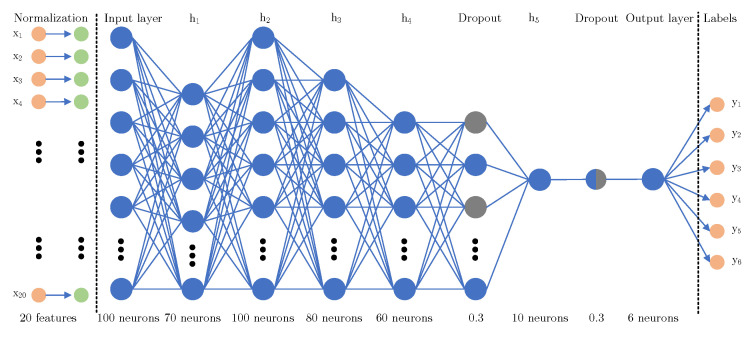
General scheme of the deep neural network structure.

**Figure 8 sensors-23-03122-f008:**
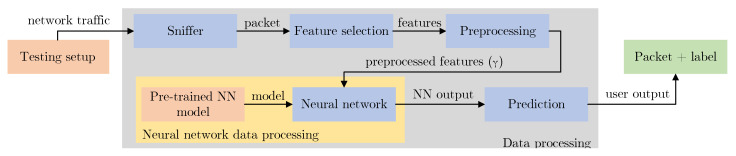
Implementation of our model in the testbed.

**Table 1 sensors-23-03122-t001:** Summary of the state-of-the-art in railway anomaly detection.

Work	Year	ML Focus	ML Methods	Area of Interest	Type of Data	Data Origin	Dataset	Real Op.
**Focused on Cybersecurity**								
[18]	2019	Supervised, Semi-supervised	RF, SVM, OCSVM, IF, DNN	electric traction substation	network comm.	physical testbed	Electra	No
[19]	2020	Supervised	ANNs	signal box	network comm.	physical testbed	Not provided	No
[20]	2022	Unsupervised	ENN, CNN, LSTM, DNN	CANbus data	network comm.	not mentioned	Not provided	No
[21]	2022	Supervised	SVM, OCSVM, RF, IF, NN, GAN+DNN	electric traction substation	network comm.	physical testbed	Electra	No
**System health, system functions, faults detecting**								
[22]	2019	Unsupervised	K-means, Self-organizing map and Autoencoders	track switches	operating data	real operation	Not provided	No
[23]	2019	Unsupervised	IF, Autoencoders	WILD, HBW, RB	operating data	real operation	Not provided	No
[24]	2021	Supervised	SVR	wheel flange lubricator	numerical data	real operation	Not provided	No
[25]	2022	Supervised	SVM	pressure sensors	signals	real operation	Not provided	No
[26]	2022	Unsupervised	SAE, MD	strain sensors	signals	real operation	Not provided	No

**Table 2 sensors-23-03122-t002:** Input data taken from captured packets from the specific layer.

Layer 0	Layer 3	Layer 4	Layer 7
Epoch time,	Length of packet	Source port	Payload
Length of frame	Length of header	Destination port	Destination address
	Identifier		Source address
	Flags		Destination port
	DS field		Source port
	Fragmentation offset		RX control byte
	TTL Size		TX control byte
	Checksum		RX timestamp
	Protocol		TX timestamp

**Table 3 sensors-23-03122-t003:** Comparison of achieved results using DNN.

Approach	No. of Test Data	Type of Label/Class	Accuracy	Precision	Recall	Specificity	F1-Score
6 labels Scenario I	35,355	Clean traffic	0.767	0.324	1.000	0.737	0.489
		Clean traffic *	0.845	0.420	1.000	0.826	0.591
		Section occupied	0.897	N/A	0.000	1.000	N/A
		DoS attack	0.922	1.000	0.467	1.000	0.637
		Replay attack	0.929	0.849	0.776	0.966	0.811
		MitM	0.921	1.000	0.490	1.000	0.658
		MitM *	1.000	0.732	1.000	1.000	0.846
		Network scanning	0.956	0.866	1.000	0.938	0.928
5 labels Scenario II	35,355	Regular traffic	0.922	0.732	1.000	0.901	0.846
		Regular traffic *	1.000	1.000	1.000	1.000	1.000
		DoS attack	0.973	1.000	0.821	1.000	0.902
		Replay attack	0.929	0.850	0.778	0.966	0.813
		MitM	0.922	1.000	0.490	1.000	0.658
		MitM *	1.000	1.000	1.000	1.000	1.000
		Network scanning	0.956	0.867	1.000	0.939	0.929
2 labels Scenario III	58,231	Regular traffic	0.920	0.869	1.000	0.831	0.930
		Attack	0.920	1.000	0.831	1.000	0.908

The * indicates cases where a combination with an active man-in-the-middle detection is considered.

**Table 4 sensors-23-03122-t004:** Comparison of the most successful algorithms of classical ML.

Algorithm	No. of Test Data	Type of Label/Class	Precision	Recall	F1-Score	Accuracy
RF	31,205	Clean traffic	1.00	1.00	1.00	1.00
		Section occupied	1.00	1.00	1.00	
		DoS attack	1.00	1.00	1.00	
		Replay attack	1.00	1.00	1.00	
		MitM	1.00	1.00	1.00	
		Network scanning	1.00	1.00	1.00	
kNN	31,205	Clean traffic	0.37	1.00	0.54	0.75
		Section occupied	0.83	0.78	0.80	
		DoS attack	0.96	0.70	0.81	
		Replay attack	1.00	0.71	0.83	
		MitM	0.83	0.73	0.78	
		Network scanning	0.95	0.72	0.82	
MLP	31,205	Clean traffic	0.00	0.00	0.00	0.50
		Section occupied	0.47	0.49	0.48	
		DoS attack	0.73	0.47	0.57	
		Replay attack	1.00	0.10	0.18	
		MitM	1.00	0.49	0.66	
		Network scanning	0.41	1.00	0.58	

**Table 5 sensors-23-03122-t005:** Results of the accuracy_score of the tested algorithms in the second phase.

Algorithm	Accuracy Score P2	Accuracy Score P3
RF	1.00	1.00
DTC	1.00	1.00
kNN	0.75	0.54
MLP	0.50	0.15
AdaBoost Classifier	0.39	0.38
Bernoulli NB	0.37	0.39
SVC	0.35	0.41
Gaussian NB	0.29	0.27

## Data Availability

Not applicable.

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
