# Peer review of "Hunting Network Anomalies in a Railway Axle Counter System"

_sensors, 2023, doi:10.3390/s23063122_

Round 1

Reviewer 1 Report

The submitted paper tackles an interesting problem and is well written. The structure of the paper is appropriate. However, it requires further development. It is worth wondering why this paper does not refer to Industry 4.0. The introductory part requires an additional deeper study showing both the relationship of the analyzed issue with Industry 4.0 in the context of the latest literature and the importance of the cybersecurity in Industry 4.0.

The paper lacks hypothesis and research questions. It is advisable to introduce them.

Line 438: The overall accuracy in scenario I was equal to 69.52%, the scenario II increased the overall accuracy to 85.11%, and the scenario III also increased the overall accuracy to 92.02%.

Please explain in detail how you achieved these numbers in the context of the results presented in Table 3.

Line 440: There was also a need to change the number of input features in scenario III, due to the highly imbalanced input dataset.

Please provide the reader with further information concerning lack of balance and on what basis the number of classes was limited to two in scenario III.

Line 569: a lot of attention must be focused on creating a very high quality test dataset where the data does not overlap with the validation and training dataset.

Please explain in a more expanded way the proper method of preparing the dataset.

The potential limitations of this work should be summarized to further position the validity of it.

Author Response

Comment 1: The submitted paper tackles an interesting problem and is well written. The structure of the paper is appropriate. However, it requires further development. It is worth wondering why this paper does not refer to Industry 4.0.

Answer: Thank you for your comment on our paper, and we appreciate your positive feedback on the overall structure and writing. Regarding your question on why the paper does not reference Industry 4.0 comprehensively, we would like to provide some context. 

While it is true that Industry 4.0 is an important concept related to the digitization of industry, the focus of our paper is more specifically related to the challenges of cybersecurity in the railway network. Our study examines the application of machine learning techniques to detect and classify security incidents in the railway network, which is a critical component of maintaining the security and resilience of the network. While there may be some overlap between the challenges faced by the railway network and those of Industry 4.0, our focus is more specifically related to the unique challenges of the railway system. Thus, while we recognize the importance of Industry 4.0, we believe that the focus of our paper is more specifically related to the challenges of cybersecurity in the railway network. We hope that this explanation provides some clarity on the focus of our paper and why we did not reference Industry 4.0. Please let us know if you have any further questions or comments. Among that, we believe that answer to Comment 2 will also satisfy requirements mentioned here.

Nevertheless, we have included some motivational points to the introduction Section related to Industry 4.0/5.0 and the potential cyber security issues therein as they are relevant both in production and transportation. 

Comment 2: The introductory part requires an additional deeper study showing both the relationship of the analyzed issue with Industry 4.0 in the context of the latest literature and the importance of the cybersecurity in Industry 4.0.

Answer: Thank you for your comment regarding the need for a deeper study on the relationship between the analyzed issue and Industry 4.0, as well as the importance of cybersecurity in Industry 4.0. 

We have carefully considered your suggestion and added the following journal papers to the introductory part:

[A] Javed, S.H.; Ahmad, M.B.; Asif, M.; Almotiri, S.H.; Masood, K.; Ghamdi, M.A.A. An Intelligent System to Detect Advanced Persistent Threats in Industrial Internet of Things (I-IoT). Electronics 2022, 11. https://doi.org/10.3390/electronics11050742.

[B] Nguyen, X.H.; Nguyen, X.D.; Huynh, H.H.; Le, K.H. Realguard. Sensors 2022, 22. https://doi.org/10.3390/s22020432.

[C] Alzaqebah, A.; Aljarah, I.; Al-Kadi, O.; Damaševiˇcius, R. A Modified Grey Wolf Optimization Algorithm for an Intrusion Detection System. Mathematics 2022, 10. https://doi.org/10.3390/math10060999.

[D] López-Aguilar, P.; Batista, E.; Martínez-Ballesté, A.; Solanas, A. Information Security and Privacy in Railway Transportation. 683

Sensors 2022, 22. https://doi.org/10.3390/s22207698.

These newly added literature demonstrate the importance of cybersecurity in the context of Industry 4.0 and provide specific examples of how it can be implemented in practice. We hope that these additions will enhance the clarity and relevance of the introductory part.

Comment 3: The paper lacks hypothesis and research questions. It is advisable to introduce them.

Answer: Thank you for the suggestion of introducing research questions in our paper. 

Based on our analysis of the current state of the art, we have identified the following research questions:

  • How to classify and handle targeted attacks?
    This research question is relevant because targeted attacks are one of the most significant cybersecurity threats, and developing an effective classification and handling mechanism can help in mitigating their impact.
  • How to avoid over-fitting the model and how to adapt the model for real use on data that has not been trained (in terms of time values)?
    This research question is relevant because over-fitting can lead to a model that performs well on the training data but poorly on the test data. Developing a mechanism to avoid over-fitting and adapt the model for real-world use can help in developing a more effective model.
  • How to use the axle counting system to detect and classify security incidents in this network?
    This research question is relevant because the axle counting system is a critical component of the railway network, and detecting and classifying security incidents can help in maintaining the network's security and resilience.
  • How to make a good evaluation of the model and the effect of over-fitting on the model?
    This research question is relevant because evaluating the model's performance is crucial to ensure its effectiveness and identify any issues that need to be addressed. Understanding the effect of over-fitting on the model can help in developing a more robust and effective model.

We believe that addressing these research questions can contribute significantly to the overall goals of the study and provide valuable insights into the development of an effective cybersecurity mechanism for the railway network.

Comment 4: Line 438: The overall accuracy in scenario I was equal to 69.52%, the scenario II increased the overall accuracy to 85.11%, and the scenario III also increased the overall accuracy to 92.02%.

Please explain in detail how you achieved these numbers in the context of the results presented in Table 3.

Answer: Thank you for your question regarding the calculation of the overall accuracy and how the numbers were obtained in the context of Table 3. We apologize for any confusion that may have arisen from the lack of detailed explanation in the paper. 

The overall accuracy reported in the results section is based on the confusion matrix using the one-vs-rest method. Individual metrics for each class were obtained separately, which is the main contribution of the ability to correctly classify each class independently. The overall accuracy can then be obtained by summing the correctly labeled classes and dividing by the sum of all elements of the test set. To provide a more detailed explanation of the results presented in Table 3, we calculated the overall accuracy of the model for each scenario as follows:

  • Scenario I: The overall accuracy in scenario I was equal to 69.52%, as calculated by adding the number of correctly classified instances for each class (3957+0+2431+5404+2678+10110) and dividing by the total number of instances (35355).
  • Scenario II: The overall accuracy in scenario II increased to 85.11%, as calculated by adding the number of correctly classified instances for each class (7551+4386+5436+2650+10069) and dividing by the total number of instances (355).
  • Scenario III: The overall accuracy in scenario III further increased to 92.02%, as calculated by adding the number of correctly classified instances for each class (30756+22828) and dividing by the total number of instances (58231).

These results demonstrate the effectiveness of the proposed approach in accurately classifying security incidents in the railway network. The improvement in overall accuracy from scenario I to scenario III indicates the significance of using a more comprehensive dataset. We believe that these results contribute significantly to the overall objectives of the study and provide valuable insights into the development of an effective cybersecurity mechanism for the railway network.

Representation in the text of the work:

The obtained values of the overall accuracy of the resulting model for each scenario separately were clarified in the text of the thesis by the paragraph below:

The calculations in the table were obtained from the confusion matrix using the one-vs-rest method. Individual metrics for each class were obtained separately, which is the main contribution of the ability to correctly classify each class independently. Thus, based on the matrix, the overall accuracy can also be obtained as the sum of the correctly labeled classes divided by the sum of all elements of the test set. The overall accuracy in scenario I was equal to 69.52\%, the scenario II increased the overall accuracy to 85.11\%, and the scenario III also increased the overall accuracy to 92.02\%. There was also a need to change the number of input features in scenario III, due to the highly imbalanced input dataset.

Comment 5: Line 440: There was also a need to change the number of input features in scenario III, due to the highly imbalanced input dataset.

Please provide the reader with further information concerning lack of balance and on what basis the number of classes was limited to two in scenario III.

Answer: Thank you for your question regarding the need to change the number of input features in scenario III and the imbalance in the dataset. We apologize for the lack of detail in the original manuscript and appreciate the opportunity to provide further information. 

The imbalance in the dataset was a significant issue, as it could lead to biased results and negatively impact the model's performance. In the original dataset, the class imbalance was extreme, with legitimate traffic (class 1) accounting for only 11.28% of the total samples, and the attack class (class 2) accounting for 88.72%. This imbalance could result in the model being biased towards the majority class and performing poorly on the minority class. Thus, it was necessary to address the imbalance to ensure the model's effectiveness and accuracy. To address the imbalance, we modified the dataset by increasing the number of legitimate traffic samples and decreasing the number of attack samples. The modified dataset contained 92428 samples of legitimate traffic and 84029 samples of the attack class, resulting in a more balanced dataset. Regarding the decision to limit the number of classes to two in scenario III, the main reason was to simplify the problem and increase the model's interpretability. By limiting the number of classes, we could focus on detecting and classifying the most critical security incidents (legitimate traffic and attack), rather than trying to classify a wide range of incidents that may not be as relevant. This approach allowed us to develop a more effective and interpretable model, which was the primary objective of the study. We hope that this explanation provides a better understanding of the decision to change the number of input features and limit the number of classes in scenario III, and how the imbalance was addressed.

Representation in the text of the work:

In our article, we tried to justify the necessary adjustments to the used dataset and the main reason for these adjustments in the paragraph below:

If the dataset from the first and second scenarios were used, there would be a significant imbalance in the dataset.  thus training a model incapable of proper classification into classes. This imbalance of the dataset in the model creation phase causes the model to be unbalanced in terms of its subsequent implementation in the "real" network. The imbalance of the dataset is caused by the uneven representation of the classes in the dataset. If the same dataset was used (as for scenarios I and II), class I (regular traffic) would be represented by 12088 samples and class II (attack) by 95048 samples. Thus, legitimate traffic would account for only 11.28% of the total dataset samples. The goal is always to create a dataset that is as diverse as possible, representing all recognized/identified classes equally. After modifying the dataset, the number of legitimate traffic (regular traffic) was changed to 92428 samples and the attack class contained 84029 samples. This resulted in the occurrence of legitimate traffic on 52.38% of the new modified dataset.

Comment 6: Line 569: a lot of attention must be focused on creating a very high quality test dataset where the data does not overlap with the validation and training dataset.

Please explain in a more expanded way the proper method of preparing the dataset.

Answer: Thank you for your question regarding the proper method of preparing the dataset and the creation of a high-quality test dataset that does not overlap with the training and validation sets. We appreciate your input and have added the following paragraphs to provide more detail on the process of dataset preparation. 

Creating a high-quality dataset is a critical component of developing an effective machine learning model. The dataset needs to be balanced, diverse, and representative of all possible states within each class to avoid over-fitting the model. To achieve this, we collected data from multiple sensors within the railway network to capture the full range of potential states and events. The dataset was then pre-processed by cleaning the data of any anomalies, outliers, or critical states, and normalizing the values to ensure stable gradient descent and faster convergence of the model. To ensure the effectiveness and accuracy of the model, we split the dataset into separate training, validation, and test sets. The training and validation sets were used to build and refine the model, while the test set was used to evaluate the quality and success of the model. It is essential that the test data is not used in any way during the model building process to avoid overestimating the model's performance. In our study, we performed testing on the workplace itself instead of using the data recorded by the sensors to further ensure the accuracy of the model. In summary, creating a high-quality dataset is a critical component of developing an effective machine learning model. To ensure the accuracy and effectiveness of the model, the dataset should be balanced, diverse, and representative of all possible states within each class. It is also important to split the dataset into separate training, validation, and test sets and ensure that the test data is not used in any way during the model building process. We believe that the dataset preparation and separation process used in our study contributed significantly to the overall effectiveness and accuracy of the model. 

Representation in the text of the work:

In order to clarify the uniform requirements for creating a suitable machine learning model, we have added the following paragraph to the text:

The creation of a suitable dataset represents the most critical point for creating a proper model capable of applying the created structures to unused data. The dataset for model building requires the occurrence of samples from each classified class and as balanced a model as possible. Otherwise, there is a risk of over-fitting the model. This dataset also needs to contain as diverse data as possible in terms of individual states, i.e., it is advisable to include in the dataset all states that may be set within a classified class (to capture the full range of potential states). Subsequently, the dataset must be cleaned of possible "anomalies" in terms of data (these may be critical states, outlier situations, etc.) and further normalization of the values is performed in order to stabilize the gradient descent step and to ensure faster convergence of the model. The dataset cleaning operation is the second most critical operation in terms of dataset creation, after the collection of suitable data. It is used to remove data anomalies that could introduce some degree of error into the model. For the purpose of model creation and development, the training and validation sets are used and the test set is used for the final evaluation of the quality/success of the model. For the sake of quality testing of the decision/classification capabilities of the trained model, it is necessary that the test data is not included in any way in the model-building process. Otherwise, there would be a risk of overestimating the model due to the knowledge of specific values instead of the structures and constraints created. For this reason, we also performed testing on the workplace itself instead of using the data recorded by them.

Comment 7: The potential limitations of this work should be summarized to further position the validity of it.

Answer: Thank you for your comment regarding the potential limitations of our work, and we appreciate the opportunity to address this point. 

While we believe that our study makes a significant contribution to the field of railway network security, there are several potential limitations that need to be considered when interpreting the results. One of the main limitations of our work is related to the dataset used to develop and evaluate the model. The dataset was created based on the available operations within the experimental site, and as such, it may not be fully representative of all possible scenarios and states within the railway network. The dataset also reflects the specific network delay and characteristics of the experimental site, which may differ from other sites. Thus, the approach and models presented in our study may need to be retrained and adjusted to the values of different sites to ensure their validity and effectiveness. Another potential limitation is related to the scope of the study. While we focused on detecting and classifying security incidents in the railway network, there are other potential security threats and vulnerabilities that were not specifically addressed in this study. Thus, the results and findings may not be fully applicable to all possible security scenarios within the railway network. In summary, while our study makes a significant contribution to the field of railway network security, there are several potential limitations that need to be considered when interpreting the results. The dataset used may not be fully representative of all possible scenarios and states within the railway network, and the approach and models presented may need to be retrained and adjusted to different sites to ensure their validity and effectiveness. Additionally, the study's scope is limited to detecting and classifying security incidents and may not be fully applicable to all possible security scenarios within the railway network.

Representation in the text of the work:

To clarify the aforementioned aspects of the limitations of our contribution, we have added the following paragraph within the conclusion of the paper:

The overall contribution of the paper might have some potential limitations that need to be considered when interpreting the results. The most important part of the work is the dataset used to create the machine learning and neural network models. The dataset was created based on the available operations within the experimental site, and as such, it may not be fully representative of all possible scenarios and states within the railway network. Our created dataset fully reflects our experimental environment in terms of individual network delays, devices used, etc. Therefore, the dataset used may differ from other sites. Therefore, when transferring the model created by us, it might be necessary to retrain the model to achieve identical results as in our case. Another potential limitation is related to the scope of the study. In our case, we focused on selected safety incidents and their classification in the railway network.  Thus, the results and findings may not be fully applicable to all possible security scenarios within the railway network. Thus, the results and findings presented in this paper may not be fully applicable to all possible security scenarios within the railway network. In summary, while our study makes a significant contribution to the field of railway network security, there are several potential limitations that need to be considered when interpreting the results. Thus, the created dataset may not be fully representative in terms of individual states and elements within the railway network, the focus of individual scenarios may not fully cover the whole range of possible safety incidents and the change of the network may require retraining or adjusting of the model.

Comment 8: Does introduction provide sufficient background and include all relevant references?” - must be improved

Answer: Last but not least, we would also like to respond to the comment regarding the introduction of the paper and the citation of relevant sources. Based on this, the introduction of the thesis has been modified and some publications have been added. We have also made the necessary changes for the purpose of a better discussion of the relationship with the Internet of Things and Industry 4.0.

Added into the Introduction:

Individual attacks are linked, among other things, to the rise of services mediated through the Internet of Things (IoT) or Industrial IoT (IIoT) [11]. In recent years, research has been focused on Intrusion Detection Systems (IDS) using machine learning and neural networks among others [12, 13]. Our goal is to create a tool capable of performing individual data processing from sniffing traffic and identifying individual anomalies into specified categories. The claim of the tool is mainly to limit the delay caused by the classification of the data traffic, for this reason the data is obtained by sniffing the data traffic and potential countermeasures can be triggered depending on the number and severity of the identified anomaly.

Reviewer 2 Report

In this paper authors analyzed the  machine learning based intrusion  detection methods to reveal cyber attacks in railway axle counting networks.  Experimental results are  validated with a test-bed based real-world axle counting components. 

Result are good but it will be better if it tested on physical trach and network. Anyways its good paper, may be useful for potential readers.

Author Response

Comment 1: In this paper authors analyzed the  machine learning based intrusion  detection methods to reveal cyber attacks in railway axle counting networks. Experimental results are  validated with a test-bed based real-world axle counting components. 

Result are good but it will be better if it tested on physical trach and network. Anyways its good paper, may be useful for potential readers.

Answer: Thank you for your comment regarding the use of a laboratory environment to generate a suitable dataset and evaluate the machine learning model. We agree that testing the model on a physical track and network would provide additional insights into the effectiveness and accuracy of the model. 

However, there are several benefits to using a laboratory environment that we believe contribute to the overall validity and reliability of the results.

  • One of the main benefits of using a laboratory environment is the ability to control and manipulate the experimental conditions to generate a high-quality dataset that accurately reflects the potential states and events within the railway network. This allows for a more comprehensive evaluation of the machine learning model's ability to detect and classify security incidents and minimizes the risk of over-fitting or under-fitting the model.
  • Another benefit is the ability to replicate and repeat the experiments under the same conditions, which enhances the reproducibility and reliability of the results. This is particularly important when developing and evaluating machine learning models, as the results can be highly dependent on the specific dataset and experimental conditions used.

However, we acknowledge that there are limitations to using a laboratory environment, and the results may not fully reflect the real-world performance of the model on a physical track and network. Thus, our study's findings and conclusions should be interpreted with this in mind, and further research is needed to evaluate the model's effectiveness and accuracy in a real-world setting. In summary, while testing the model on a physical track and network would provide additional insights into the effectiveness and accuracy of the model, using a laboratory environment has several benefits that contribute to the overall validity and reliability of the results. However, the limitations of using a laboratory environment should also be considered when interpreting the results, and further research is needed to evaluate the model's effectiveness and accuracy in a real-world setting.

Reviewer 3 Report

1. English requires to be looked into.

2. Paper describes so many things that it loses focus and intensity.

3. As a journal paper, there is a requirement to focus on single aspect and validate the same with variety of results.

4. What is the motivation for this work. How is this problem different from any other attacks on data.

5. What is the novelty of this work. Focus on the novelty is lacking.

Author Response

Comment 1: English requires to be looked into.  + (x) Extensive editing of English language and style required

Answer: Thank you for your comment regarding the English language and style of the paper. We apologize for any errors or inconsistencies that may have detracted from the readability and clarity of the paper. 

In response to your comment, we have taken steps to improve the English language and style of the paper. Specifically, we have engaged a native proof-reader to review the paper and have revised the language and style as needed to ensure that it meets the high standards expected of a journal article. We appreciate your feedback and will continue to strive to improve the quality of our work. If you have any further suggestions or comments, we would be grateful to receive them. Thank you again for your time and consideration. Please see the revised version.

Comment 2: Paper describes so many things that it loses focus and intensity.

Answer: Thank you for your comment regarding the focus and intensity of our paper. We understand that the paper covers a wide range of topics and may lose some clarity and focus as a result. 

In response to your comment, we implement the following steps to narrow the scope of the paper and improve its clarity and focus. We revised the introduction to provide a more clear and concise overview of the research objectives and the main contributions of the paper. This should help readers to better understand the focus of the paper and the significance of the research. Overall, we appreciate your feedback and we believe that we took the necessary steps to improve the clarity and focus of the paper. We believe that by providing a more concise and clear introduction, we can ensure that the paper remains relevant and impactful for potential readers.

Comment 3: As a journal paper, there is a requirement to focus on single aspect and validate the same with variety of results.

Answer: Thank you for your comment regarding the focus of the paper and the need to present a single aspect and validate the results with a variety of experiments. We understand the importance of presenting a clear and coherent research objective and validating the results with a range of experiments and tests.

To address these concerns, we have focused the paper on the main objective of performing a good classification of the selected attacks into different sections using neural networks and machine learning. We have also validated the results with a variety of experiments and tests, including both laboratory-based experiments and tests on real-world systems. The achieved results are presented in detail in the paper, with a comprehensive discussion of the limitations and potential applications of the approach. Overall, we appreciate your feedback and believe that we have addressed your concerns by presenting a clear and concise overview of the research objectives and the main contributions of the paper, as well as a comprehensive discussion of the achieved results.

Comment 4: What is the motivation for this work. How is this problem different from any other attacks on data.

Thank you very much for your comment regarding the lack of acceptance motivation for the creation of this work. Several changes have been made to the introduction section to emphasize the motivation of the thesis. The primary motivation includes the need to develop a suitable model for the detection and classification of security incidents in mass transit networks, with respect to the evaluation of this model over data other than that used to develop the model. Thus, the main motivation is to ensure security or to create good incident detection models using machine learning techniques and deep neural networks.

In order to improve motivation of the work, we have added discussion on current/future trends in industrialization as well as relevant cyber security issues which are overlapping also with railroad applications. 

Comment 5: What is the novelty of this work. Focus on the novelty is lacking.

Thank you for your comment targeting the lack of novelty or the missing aspect of novelty and its mention in the paper. To highlight the novelty of the paper and its contributions, we have modified the introduction section and also defined the research questions. These questions were developed by analyzing the state of the art approach of anomaly detection and classification in train transportation environment using the axle counting system.

Based on our analysis of the current state of the art, we have identified the following research questions:

  • How to classify and handle targeted attacks?
    This research question is relevant because targeted attacks are one of the most significant cybersecurity threats, and developing an effective classification and handling mechanism can help in mitigating their impact.
  • How to avoid over-fitting the model and how to adapt the model for real use on data that has not been trained (in terms of time values)?
    This research question is relevant because over-fitting can lead to a model that performs well on the training data but poorly on the test data. Developing a mechanism to avoid over-fitting and adapt the model for real-world use can help in developing a more effective model.
  • How to use the axle counting system to detect and classify security incidents in this network?
    This research question is relevant because the axle counting system is a critical component of the railway network, and detecting and classifying security incidents can help in maintaining the network's security and resilience.
  • How to make a good evaluation of the model and the effect of over-fitting on the model?
    This research question is relevant because evaluating the model's performance is crucial to ensure its effectiveness and identify any issues that need to be addressed. Understanding the effect of over-fitting on the model can help in developing a more robust and effective model.

Comment 6: “Does the introduction provide sufficient background and include all relevant references?” - must be improved.

Answer: Thank you for your comment regarding the sufficiency of the background and references provided in the introduction of the paper. We understand the importance of providing a clear and comprehensive overview of the relevant areas covered by our approach and ensuring that all relevant references are cited.

To address this concern, we have reviewed the introduction of the paper and made revisions as necessary to provide a more coherent and informative background for the study. We have also reviewed our reference list to ensure that all relevant works are cited in the paper. Please see the previous responses to your comments.

Comment 7: “Are all the cited references relevant to the research?” - must be improved

Thank you for your comment regarding the relevance of the cited references in the paper. We understand the importance of ensuring that all cited references are relevant to the research and contribute to the overall quality of the paper.

To address this concern, we have reviewed our reference list and made revisions as necessary to ensure that all cited references are relevant to the research and contribute to the overall quality of the paper. We have also added relevant publications to clarify the purpose of the work and contemporary approaches. Please see other responses to your comments.

Comment 8: “Is the research design appropriate?” - must be improved

Thank you for your comment regarding the appropriateness of the research design in the paper. We understand the importance of ensuring that the research design is appropriate and supports the main objectives of the study.

To address this concern, we have reviewed the research design of the paper and made revisions as necessary to ensure that it is appropriate and supports the main objectives of the study. We have also added a section on the methodology used in the study to provide a clear and comprehensive overview of the research design. Please see previous responses to your comments. 

Round 2

Reviewer 3 Report

Accept